# Nurses’ Perceptions on the Implementation of a Safe Drug Administration Protocol and Its Effect on Error Notification

**DOI:** 10.3390/ijerph18073718

**Published:** 2021-04-02

**Authors:** Francisco Miguel Escandell-Rico, Juana Perpiñá-Galvañ, Lucía Pérez-Fernández, Ángela Sanjuán-Quiles, Piedras Albas Gómez-Beltrán, Juan Diego Ramos-Pichardo

**Affiliations:** 1Nursing Department, Alicante University, 03690 Alicante, Spain; francisco.escandell@ua.es (F.M.E.-R.); angela.sanjuan@ua.es (Á.S.-Q.); 2Almoradí Health Center, Health Department 21 Orihuela SNS, 03300 Alicante, Spain; perez_lucfer@gva.es; 3Nursing Department, University of Huelva, 21007 Huelva, Spain; alba.gomez@denf.uhu.es (P.A.G.-B.); juan.ramos@denf.uhu.es (J.D.R.-P.)

**Keywords:** adverse events, protocol, patient safety, quality of care, medications

## Abstract

Patient safety and quality of care are fundamental pillars in the health policies of various governments and international organizations. The purpose of this study is to evaluate nurses’ perceptions on the degree of implementation of a protocol for the standardization of care and to measure its influence on notification of adverse events related to the administration of medications. This comparative study used data obtained from questionnaires completed by 180 nurses from medical and surgical units. Our analyses included analysis of variance and regression models. We observe that the responses changed unevenly over time in each group, finding significant differences in all comparisons. The mean response rating was increased at 6 months in the intervention group, and this level was maintained at 12 months. With the new protocol, a total of 246 adverse events and 481 incidents without harm was reported. Thus, actions such as the use of protocols and event notification systems should be implemented to improve quality of care and patient safety.

## 1. Introduction

Patient safety is one of the highest priorities in the health policies of various governments and international organizations [1]. In 2004, the World Health Organization (WHO) created the Alliance for Patient Safety to promote actions, tools, and recommendations designed to improve patient safety during healthcare administration. Currently, worldwide, up to four out of ten patients suffer damage in primary and outpatient health care, of which 80% can be prevented [1]. The most damaging errors are related to the prescription and use of medications [2,3]. In high-income countries, an estimated one in ten patients is harmed while receiving hospital care [4]. Similarly, in the 37 countries of the Organization for Economic Cooperation and Development (OECD), 15% of total hospital activity and expenditure is a direct result of adverse events (AEs) [4]. Other studies carried out in Australia and England [2] and in Holland [3] refer to enhancing responsibility and security and increasing notifications with better training, education, and implementation of a protocol. In fact, in 2017, the WHO established safety in the administration of medicines as the next global challenge for patient safety [5].

In Spain, the 2015–2020 Spanish Strategy for Safety and Health at Work was developed, which recommends improving the culture of safety in health organizations and incorporating health risk management tools and training professionals in basic aspects of safety [6], including those related to the prevention of medication errors (MEs). MEs are difficult to prevent before they reach the patient and can have adverse consequences both in terms of morbidity and mortality [7,8,9,10,11,12]; they are the result of a multitude of factors, including human errors as well as characteristics of the health system or institutions themselves, such as the availability of resources, care overload, distractions, anxiety in professionals, and a lack of standardized protocols for the preparation and administration of medications [13,14,15].

The recommendations from the International Alliance of the WHO for Patient Safety and the National Health System in Spain (2015–2020) [6] were specifically designed to reduce MEs, including the use of notification systems for improving the “culture of safety in hospitals” and the implementation of action protocols for the preparation and administration of drugs in their considerations [16,17]. Protocols support health professionals by describing processes and interventions, and compliance with them helps reduce errors [18,19,20,21,22,23,24].

The objective of this study is to evaluate nurses’ perceptions on the degree of implementation of a protocol for standardization of the preparation and administration of medications, and its influence on the notification of AEs related to MEs in a General University Hospital in the Spanish National Health System.

## 2. Materials and Methods

### 2.1. Design and Study Subjects

The type of study was quasi-experimental non-randomized, and the study was carried out at the General University Hospital of Elda (Alicante-Spain). The organization of hospital services is divided into two independent buildings corresponding to medical units (internal medicine and medical specialties) and surgical units (traumatology and orthopedics, surgery, and gynecology/obstetrics), a situation that allowed for separation of the intervention and control groups, respectively. A gradual approach was made toward the development of a complex intervention: a protocol with the objective of standardizing and administering the medication prescribed to the patient in conditions of safety and quality. It was initially implemented in medical units, and during this process, the nursing professionals in these units received specific training on the protocol during the implementation process. Participant data were collected at the beginning of the implementation process and during a 12-month follow-up period. As a comparison group, data were collected from nurses in surgical units in which implementation of the protocol had not yet begun. The inclusion criteria in both groups were more than two years of experience working in the unit and having a current employment contract with more than 12 months until its expiration. Nurses who explicitly expressed not wanting to participate or did not sign the informed consent were excluded.

### 2.2. Protocol Development

During the development of the new protocol, we reviewed the current scientific literature and national reference documents on drug administration safety. We consulted the Patient Safety Strategy (2015–2020) [6] and Good Practice Guide for the Preparation of Medicines (GBPP) [25] from the National Health System, the report on the evolution of the implementation of safe drug use practices in Spanish hospitals (2007–2011) [26], and the current protocols in use at different Spanish hospitals [27,28,29]. The Web of Science, Scopus, and PubMed databases were consulted to search for academic literature by combining the following keywords “nursing”, “protocol”, “medication”, and “safety”.

Once the relevant information was located, the protocol used in this study was developed by using the Delphi methodology to help us reach a consensus among a group of experts. For the selection of experts, healthcare professionals directly related to management fields (experts in safety and quality), teaching, research, and healthcare provision were considered. Thus, we selected 10 expert professionals in advanced-practice nursing in different fields in order to ensure the plurality of the approaches examined.

The Delphi study was divided into two phases:Phase I. In this stage, we consulted the annual report on the notification of adverse drug events at the Elda hospital [30] and aimed to agree upon the skills that healthcare nurses require to safely administer medications. This aspect was carried out in two rounds of the Delphi process (definition and formation of the group of informants) with the panel of experts.Phase II. This second phase was designed to agree upon the development of each procedural skill for the safe administration of drugs through the protocol as well as the type of training required to achieve these respective levels of development. Round 3 included the execution of consultations, and in round 4, the results of the Delphi process were implemented for this phase.

For the purpose of this research, “consensus” was defined as a minimum of 80% convergence in the opinions of the experts.

To comply with the ethical standards related to human experimentation, we submitted our study protocol for consultation and approval by the Nursing Procedures Commission at the Elda Health Department. The commission comprised 10 nurses who were experts in the advanced practice of nursing procedures in the hospital setting (who differed from those participating in the Delphi group). The protocol was approved on 8 March 2018.

### 2.3. Implementation Process

Up until the development and implementation of the new protocol described in this article, Elda Hospital followed the guidelines developed for the process of safe administration of medicines collected in the Nursing Activities Guide published by the Universal Ministry of Health and Public Health of the Valencian Community (Conselleria de Sanitat Universal i Salut Pública de la Comunitat Valenciana; 2011) [31]. The implementation process of the new protocol was piloted in the hospital’s medical units, while the previous guidelines were maintained in the surgical and critical care units.

The nurses in the units where the new protocol was applied completed training sessions prior to the new protocol implementation (intervention group; IG) and were followed up at 0, 6, and 12 months in order to increase their awareness of the importance of standardization and of updating the protocols for the safe administration of medicines procedures. According to these guidelines as well as the standards of safety and quality, the sessions (1) presented the new protocol, (2) defined its legal framework and current general guidelines for the procedures applied for the safe administration of drugs, (3) presented the computer systems used for the notification of AEs, (4) and explained the concepts of clinical safety. This training was subsequently accredited and recognized by the Commission for Ongoing Training of Spanish National Health System Health Professionals in the Valencian Community (Comisión de Formación Continuada de los Profesionales Sanitarios de la Comunidad Valenciana del Sistema Nacional de Salud de España).

### 2.4. Variables

We evaluated the nurses’ perceptions on the degree of implementation of the new protocol, relating it to MEs in the context of notification of AEs. To achieve this, we used the Safety of the System for the Use of Medicines in Hospitals Self-Assessment Questionnaire, adapted from the Institute for Safe Medication Practices (ISMP) Medication Safety Self-Assessment^®^ for Hospitals [32], which was first validated and authorized by the Institute for Safe Medication Practices (ISMP) in Spain. Thus, considering the patient quality and safety standards provided in the National Health System Patient Safety Strategy for the 2015–2020 period, strategic line 2: “Safe clinical practices” [6], and after reaching a consensus among our research group, we selected 10 questions that reflect the factors we wanted to address in this current study.

The final questionnaire we used had a Likert-type response scale with 5 options, as follows: 0, “No efforts have been made to implement this point.”; 1, “This point has been discussed for possible implementation but has not yet been implemented.”; 2, “This point has been partially implemented in some or all areas of the institution.”; 3, “This point has been fully implemented in some areas of the institution.”; and 4, “This point has been fully implemented throughout the institution.” Thus, the maximum possible score for the questionnaire was 40 points, which therefore represented full implementation of all the practices.

The evaluation was carried out at four different time points: at the starting pre- and post-intervention phases of the protocol implementation (T0), at 6 months (T6), and at 12 months (T12), and the overall score was calculated as the sum of the scores for the 10 items on the questionnaire. In addition, the following sociodemographic data were collected for all participants: age, sex, number of years since the completion of their academic training, and the number of years of experience in the unit in which they worked.

The data regarding the notification of AEs were obtained through the annual report on the notification of AEs, prepared by the hospital’s risk management working group during the pre- and post-implementation periods of the new protocol. These data were extracted through the Adverse Event Notification System (SINEA in its Spanish acronym), which was created to collect and channel information on patient safety events occurring in the Valencian Community (Generalitat Valenciana, Conselleria de Sanitat, SINEA, 2013) [30].

### 2.5. Ethical Considerations

After receiving approval from the Clinical Research Ethics Committee (code ADMED 1.0), we provided the nurses in the medical units with an information sheet about this study and requested their informed consent to participate. This study complied with the ethical principles for medical research on human beings established in the Declaration of Helsinki. The data were considered strictly confidential and were protected from unauthorized use by anyone other than the study authors. The confidentiality of the participants was also respected during the data processing and analysis. At all times, the participants were identified with their own study code so that they all carried the acronym ADMED and then three numbers in sequence (ADMED001, ADMED002, etc.).

### 2.6. Data Analysis

We performed a descriptive analysis of all the variables by calculating their frequencies. Contingency tables were used to analyze the homogeneity of the control group (CG) and IG, applying Chi-squared tests for the categorical variables and by comparing the mean values for the quantitative variables using Student *t*-tests. Linear regression models were adjusted to the responses at each time point to analyze their association with the covariates and group factors. A repeated measures analysis of variance procedure was performed to observe the changes in response over time in each group. The programs used were Excel for the database and SPSS software (version 25.0; IBM Corp., Armonk, NY, USA), and the results were considered significant if *p* < 0.05.

## 3. Results

A total of 180 nurses participated in this study, with 90 each in the IG and CG. In relation to the sociodemographic variables, 88.9% were women and 74.4% had attended the University of Alicante (Spain). The participants had completed their training a mean of 16.09 years prior and had been providing care in their unit for an average of 10.81 years.

The highest mean questionnaire item scores were obtained for questions related to current protocols (item 1) and the provision of information to patients and relatives (item 10). The lowest mean scores were obtained for the questions about separated drug preparation areas (item 4), distractions and noises (item 5), and staffing (item 6). The average scores increased in the IG between 0 and 6 months and remained stable from 6 to 12 months.

Table 1 shows the means and standard deviations (*SD*s) for each of the survey questions in both the CG and IG during different phases of the study.

Table 2 shows a comparison of the data by age, years since completion of training, and years of work experience. No differences were observed between the IG and CG for any of these variables.

We observed significant differences in all the comparisons when we analyzed the mean scores in the questionnaire at the different measurement time points (Table 3).

Figure 1 shows how the responses changed unequally in each group over time. When the intervention was first implemented (T0), the response distribution was similar in both groups. However, compared to the CG, the response was greater in the IG at both 6 and 12 months. We observed that the mean scores at each time point did not differ over time for the CG while the mean response for the IG significantly increased at 6 months and remained stable between 6 and 12 months.

Data from the annual report on the notification of AEs indicated that, during use of the old protocol in 2018, 173 AEs (falls, MEs, allergic drug reactions, and phlebitis) and 210 incidents without harm (misidentification, MEs, peripheral line extravasation, allergic drug reactions, and falls) were reported while, during the use of the new protocol in 2019, 246 AEs and 481 incidents without harm were reported. As shown in Table 4, the number of AEs and incidents without harm related to MEs reported before and after implementation of the new protocol increased from 2018 to 2019.

## 4. Discussion

The objective of this study was to assess nurses’ perceptions after implementing a new protocol for the safe administration of medications and to analyze its impact on systems for reporting AEs. Realization of the protocol and its subsequent implementation in the hospital in our study allowed us to reach a consensus for each criteria in each of the procedures for the safe administration of medications through the Delphi methodology and to know the type of educational intervention to achieve this level of development. We emphasize that, in the process of safe administration of medicines, nurses begin participated from the beginning of the program (prescription, preparation, dispensing, administration, registration, and evaluation). Our data shows that the number of notifications increased after implementation of the protocol, perhaps because the safety culture had improved after completion of the training sessions and protocol execution. Notification depended not only on awareness of the error but also on the good will of the staff documenting it and, especially, the climate of trust transmitted by the organization’s leaders. Creating this climate meant that notifications would be understood as opportunities to improve safety and not as a mechanism for blame or punishment [3,23,24].

This study addressed a strategy for improving the management of clinical practice because the use of protocols can help to standardize and simplify the management and administration of medications [13,16,20,22,33]. Thus, protocolization is one of the most widely used strategies to reduce both variability in clinical practice and the incidence of errors in the medication administration process [18,20,34,35,36,37,38,39].

Complete protocol implementation was reported in the IG healthcare units in terms of the information received by patients. During the administration of drugs, patients and/or relatives were not usually informed about the generic and/or commercial name of the drugs used or their action, doses, or the most prominent possible AEs. The provision of this information is an essential part of the medication administration process, although we must emphasize that this function must be carried out with great professional responsibility and while considering the related medical ethics [40,41].

In contrast, the responses from both groups remained unchanged during the pre- and post-intervention phases for the questions that referred to distractions and separated areas. In this sense, medications are often administered in difficult conditions, which can create opportunities for interruptions and distractions [14,42].

In this study, the lowest scores were obtained in reference to staffing. Here, it is important to mention the role played by health organizations because it is their responsibility to implement measures and barriers to prevent AEs, especially those with serious consequences that have already occurred at least once. Past studies that have analyzed the safety culture of organizations and measured the impact of these protocols have shown that inadequate involvement and staffing by professionals risks the safety of patients and reduces the success rates of drug administration quality plans [43]. Thus, the role of managers and staffing levels in healthcare units is crucial to achieve a positive culture of safety in healthcare institutions [1].

Regarding the training that nurses had received on medication administration systems (e.g., infusion pumps, automatic preparation equipment, etc.), partial implementation was observed in the IG units. Various authors and international organizations advise creating strategies to change the design of nursing jobs as well as the use of targeted technologies to reduce errors [44,45,46,47]. In this current study, we observed that prior training of the staff in units where a new protocol for the safe administration of medicines would be implemented helped to improve and sustain the perception that these professionals had over time on the degree of implementation and compliance with the protocol as well as its perceived usefulness. We were unable to find other studies that have analyzed this question over time.

### Limitations and Solutions

Among the main limitations of this research are those inherent to this type of study: obtaining the sample from a single hospital center and being based on the willingness of the nurses to participate. The difficulties encountered in a Delphi study are the imposition of preconceptions on the experts and the techniques of summarizing and presenting the responses of the coordinating group.

## 5. Conclusions

At a practical level, our results indicate, in the IG, nurses’ perceptions on improvements in the degree of implementation of a protocol for the safe administration of medications. We consider our contributions to be of special interest, since the protocol encourages safe clinical practices and the non-punitive characteristics of adverse event notification systems, improving the safety culture. In this way, through our study, we provided a solution in line with the action strategies proposed by the WHO and different international organizations to be a global challenge for patient safety.

## Figures and Tables

**Figure 1 ijerph-18-03718-f001:**
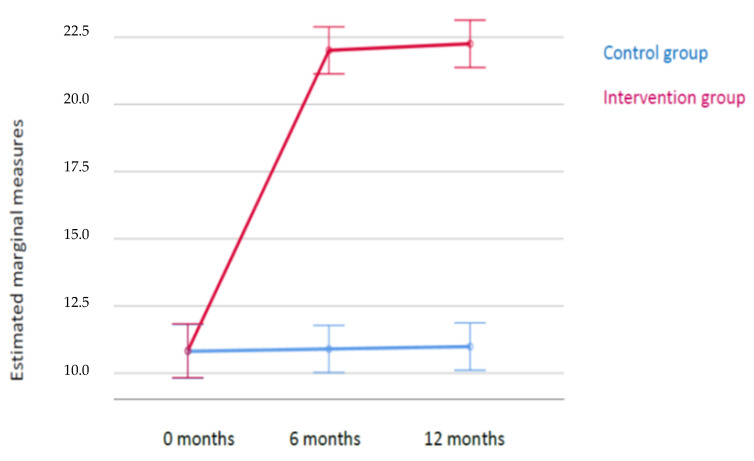
Response changes over time.

**Table 1 ijerph-18-03718-t001:** Means of the questionnaire items in the different phases of the study.

Questions		0 Months				6 Months		12 Months	
		IG PRE	CG PRE	IG POST	CG POST	IG	CG	IG	CG
C1	Mean	0.387	0.389	3.146	0.389	3.244	0.389	3.244	0.416
	SD	0.510	0.513	0.441	0.513	0.504	0.513	0.504	0.580
C2	Mean	1.133	1.178	2.567	1.178	2.633	1.178	2.678	1.169
	SD	1.212	1.214	0.720	1.214	0.800	1.214	0.846	1.208
C3	Mean	0.956	0.956	2.622	0.956	2.756	0.956	2.811	0.966
	SD	1.340	1.340	0.907	1.340	0.987	1.340	1.027	1.344
C4	Mean	1.956	1.900	1.511	1.900	1.511	1.900	1.511	1.888
	SD	1.872	1.861	0.877	1.871	0.877	1.861	0.877	1.867
C5	Mean	1.878	1.789	1.511	1.789	1.511	1.800	1.533	1.820
	SD	1.895	1.887	0.877	1.887	0.877	1.885	0.889	1.886
C6	Mean	0.156	0.156	1.178	0.156	1.178	0.156	1.178	0.157
	SD	0.539	0.539	0.384	0.539	0.384	0.539	0.384	0.541
C7	Mean	1.078	1.111	2.044	1.111	2.044	1.133	2.056	1.157
	SD	1.183	1.194	0.686	1.194	0.686	1.201	0.693	1.186
C8	Mean	0.389	0.389	1.956	0.389	1.956	0.444	1.978	0.449
	SD	0.490	0.490	0.847	0.490	0.847	0.602	0.861	0.603
C9	Mean	0.556	0.578	1.956	0.578	1.956	0.578	1.989	0.584
	SD	0.721	0.734	0.847	0.734	0.847	0.734	0.868	0.736
C10	Mean	2.356	2.367	3.133	2.367	3.222	2.367	3.278	2.393
	SD	1.819	1.826	0.545	1.826	0.595	1.826	0.619	1.819

**Table 2 ijerph-18-03718-t002:** Sociodemographic data.

		N	Mean	SD	*p*-Value
Age	Control	90	36.59	5.86	0.797
	Intervention	90	36.82	6.28	
Years since training completion	Control	90	16.12	6.15	0.942
	Intervention	90	16.06	6.17	
Experience	Control	90	10.32	4.2	0.155
	Intervention	90	11.29	4.87	

**Table 3 ijerph-18-03718-t003:** Difference in the means.

		Difference in the Means	Error	*p*-Value
Control	T0–6 months	−0.089	0.030	<0.012
	T0–12 months	−0.178	0.043	<0.000
	6 months–12 months	−0.089	0.030	<0.012
Intervention	T0–6 months	−11.178	0.623	<0.000
	T0–12 months	−11.422	0.637	<0.000
	6 months–12 months	−0.244	0.053	<0.000

**Table 4 ijerph-18-03718-t004:** Adverse events and incidents without harm.

Error Type		2018		2019	
		*n*	%	*n*	%
Incident without harm	Medication error Allergic reactions Others	58 2 150	29 1 70	83 3 395	17 0.6 82.4
Total		210	100	481	100
Adverse events	Medication error Allergic reactions Others	16 19 138	9.2 11 79.8	29 17 200	12 7 81
Total		173	100	246	100

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
