# Peer review of "Nurses’ Perceptions on the Implementation of a Safe Drug Administration Protocol and Its Effect on Error Notification"

_ijerph, 2021, doi:10.3390/ijerph18073718_

Round 1

Reviewer 1 Report

Specifically, the purpose of this study was to evaluate nurses’ perception of the degree of implementation of a protocol for the standardization of care and to measure its influence on the notification of adverse events related to the administration of medications.

This comparative study used data obtained from questionnaires completed by 180 nurses from medical and surgical units. 

It is an interesting topic, as the use of protocols  should be implemented to improve quality of care and patient safety. In this  study, it was proved that the prior training of the staff in units where a new protocol for the safe administration of medicines would be implemented, helped to improve and sustain the perception these professionals had over time of the degree of implementation and compliance with the protocol, as well as its perceived usefulness. A very important issue that  other studies  have not addressed previously.  I believe that the conclusions are consistent with the evidence and arguments presented as the authors revealed that nurses’ perceptions of improvements in the degree of implementation of the protocol for the safe administration of medications, as well as its influence on the notification of AEs in the administration of drugs, was satisfactory over an extended period.  

Author Response

Many thanks to the reviewers and editors for their time and comments. 

Reviewer 1 makes no comment. It looks good.

Thank you very much.

Reviewer 2 Report

Please review how the article was written in LaTex as some of the author guidelines are not fulfilled.

Background:

  • Background is adequate but could have a better flow, specifically in the final paragraphs, in which some sentences do not connect.

Methods:

  • The type of study is not correctly detailed (e.g. experimental/quasi-experimental);
  • The development of the protocol serves a different objective. To add this to the article, I would recommend changing the type of study (e.g. stepwise approach for the development of a complex intervention - protocol);
  • To add the variable sex to the analysis appears to be sexism. It is a dangerous option and discriminates. Please consider removing from the inferential analysis. The same applies to the university. When choosing a variable for inferential testing there must be a clear logic to how it could influence the results.
  • The development of a conditioned latent growth model could be a better statistical option. Is it possible to develop such a model? Were the test assumptions met (e.g., normality of the residuals). The same applies to the other parametric tests.

Results

  • As detailed above, consider excluding the use of the chi-squared test in table 3.
  • With the exception of C4 and C5, the T0 scores for the IG appear to have been collected after the protocol implementation, as the difference between the IG and the CG is enormous. This is not so clear in the methodology.

Discussion:

  • As medication may be different in terms of context (medical wards vs surgical wards), could this be a limitation? If not, what is the inherent reasoning?
  • If the development of the protocol is to be included, please include it in the discussion.

Conclusion

  • The conclusion is poor and should connect with a broader and international objective.

Author Response

Many thanks to the reviewers and editors for their time and comments. The comments have been answered in the following document and modified in the text.

Reviewer 3 Report

Thank you for the opportunity to review this manuscript submission.

I want to congratulate the authors on the subject of the manuscript. In my opinion,

I found the study to be very interesting and the methods strong, and i think The topic is very interesting and is needed.

The manuscript is written in an understandable way and contains in each section the most relevant aspects of the research.

However, the manuscript need to do some major revisions.

In the Introductuion,

What is going on in the world? I only find information from Spain

In the methodology; IN the inclusion criteria, how did professionals with a seniority of more than two years include the study? They excluded nurses even though they were in the control group. I do not understand this clarification.

In the discussion it is recommended to use scientific literature more current, because there are many articles published by what we encourage them to perform a search of articles. I recommend to deepen the limitations of the study. 

Author Response

(The authors gave the same response as above.)

Reviewer 4 Report

Comments for Authors

Thank you for the opportunity to review the article entitled “Nurses’ perceptions of the implementation of a safe drug administration protocol and its effect on error notification”.

Suggestions for Authors before publication:

Abstract: There is no information on the statistical methods.

The introduction:

 Is too general and relates to general procedures for ensuring safe care for patients.

Due to the fact that it is an international journal, it is worth including examples of other countries in this area.

What is more, there is no information about the autonomy of the nursing profession in Spain, do they only carry out doctor's orders in their work, or can they carry out the procedures themselves, e.g. prescribe medicines?

This is also worth developing in the discussion.

The aim of the study is not clear and requires clarification

Methodology section:

Please provide the criteria for selecting the place of the research and the basis of the recruitment of participants.

In the limitations, the authors write that the participants were people willing to participate in the study.

In my opinion, always willing participants take part in the study.

How data was collected and how data confidentiality was ensured?

Why did the Authors not describe the various stages of Delphi method? Please complete

In addition, the limitations of the study, such as those resulting from the specificity of the Delphi study, should be supplemented - a long period of data acquisition, especially in the paper version.

Tables:

For easier data analysis, please explain all abbreviations used under the tables.

Author Response

(The authors gave the same response as above.)

Round 2

Reviewer 2 Report

Dear authors.

Overall the present manuscript answers most of the concerns of the previous review.

  • Add reference to the new sentence "of which 80% can be prevented".
  • Rephrase "refer to enhance responsibility, security and increase of notifications with greater training, education and implementation of a protocol". Perhaps only changing from "enhance" to "enhanced" will suffice.
  • In subchapter 2.1, please refer to what the protocol targets, as the term is too vague.
  • Table 1, change Media to Mean and DT to SD. Correct the decimal points according to the journal's requirements. The table from manuscprit v1 was correct.
  • Table 2 should be rearranged.
  • Chapter 4 - change "The development of the protocol of our study, it allowed us to..." to "The development of the protocol of our study allowed us to..."
  • The manuscript could benefit from an English editing service as some of the text (specifically the new introductions) are hard to read.

Author Response

(The authors gave the same response as above.)

Reviewer 3 Report

Consider that this article is not ready to be published. Important aspects are missing. In the tables it is recommended to unify the periods (.) And the commas (,) since each time it appears in one way

Author Response

Many thanks to the reviewers and editor for their time and comments. The comments have been answered in the following document and modified in the text.
